# Mitochondrial Activity and Skeletal Muscle Insulin Resistance in Kidney Disease

**DOI:** 10.3390/ijms20112751

**Published:** 2019-06-05

**Authors:** Jane E. Carré, Charles Affourtit

**Affiliations:** School of Biomedical Sciences, University of Plymouth, Plymouth PL6 8BU, UK; jane.carre@plymouth.ac.uk

**Keywords:** bioenergetics, energy metabolism, ATP turnover, oxidative stress, insulin signalling, obesity, diabetic nephropathy, uraemic myopathy, renal sarcopenia, muscle wasting

## Abstract

Insulin resistance is a key feature of the metabolic syndrome, a cluster of medical disorders that together increase the chance of developing type 2 diabetes and cardiovascular disease. In turn, type 2 diabetes may cause complications such as diabetic kidney disease (DKD). Obesity is a major risk factor for developing systemic insulin resistance, and skeletal muscle is the first tissue in susceptible individuals to lose its insulin responsiveness. Interestingly, lean individuals are not immune to insulin resistance either. Non-obese, non-diabetic subjects with chronic kidney disease (CKD), for example, exhibit insulin resistance at the very onset of CKD, even before clinical symptoms of renal failure are clear. This uraemic insulin resistance contributes to the muscle weakness and muscle wasting that many CKD patients face, especially during the later stages of the disease. Bioenergetic failure has been associated with the loss of skeletal muscle insulin sensitivity in obesity and uraemia, as well as in the development of kidney disease and its sarcopenic complications. In this mini review, we evaluate how mitochondrial activity of different renal cell types changes during DKD progression, and discuss the controversial role of oxidative stress and mitochondrial reactive oxygen species in DKD. We also compare the involvement of skeletal muscle mitochondria in uraemic and obesity-related muscle insulin resistance.

## 1. Introduction

When the blood glucose concentration rises above its set point, pancreatic beta cells secrete insulin [1], a peptide hormone that provokes an orchestrated anabolic response to the raised glucose availability by altering the behaviour of multiple organs [2]. Skeletal muscle is a major contributor to the systemic anabolic response, as insulin-sensitive glucose uptake by this comparably large tissue accounts for more than 70% of the whole-body glucose disposal [3]. Most of the glucose that is taken up by skeletal muscle is subsequently stored as glycogen, as insulin activates glycogenesis and inhibits glycogenolysis [2]. Insulin also enhances cell growth and differentiation [4], stimulates protein synthesis and inhibits protein breakdown [5,6] and promotes mitochondrial biogenesis [7] (Figure 1). When higher than usual insulin levels are required to maintain glucose homeostasis, individuals are considered to be insulin resistant. Obesity (body mass index ≥30 kg/m^2^) is a broadly accepted risk factor of systemic insulin resistance [8], and loss of skeletal muscle insulin sensitivity is the first sign of such resistance in human subjects [2]. Pancreatic beta cells compensate by increasing the production and secretion of insulin [9], but when such compensation fails, insulin resistance will progress to type 2 diabetes (T2D). In turn, T2D may cause various complications [10], including diabetic kidney disease (DKD—[11]). It has been estimated [12] that 609 million adults were obese in 2015, which amounts to roughly 10% of the global population. Equally staggering, 8.2% of adults worldwide have been diagnosed with T2D [13], 20–40% of whom are expected to develop DKD [14]. A better mechanistic insight into these disorders is necessary to curb their high prevalence.

The many proposed mechanisms by which different organs lose their insulin sensitivity are diverse, but nutrient excess is a common factor [2]. Such excess emerges when dietary nutrient supply outweighs systemic energy expenditure, and is thus a key feature of obesity. At the cellular level, mitochondria are crucial for adjusting nutrient-fuelled ATP supply to energy expenditure, and changes in oxidative ATP synthesis are linked to obesity-related insulin resistance of skeletal muscle [16]. Skeletal muscle insulin resistance also emerges in other states than obesity. For instance, individuals with chronic kidney disease (CKD) exhibit low muscle insulin sensitivity even at the very onset of renal dysfunction [17]. This uraemic muscle insulin resistance is linked to disturbed protein metabolism [5], and to the loss of skeletal muscle function and mass [18]. The mean global prevalence of CKD is currently 13.4% [19]. As in obesity, muscle insulin resistance in CKD is associated with altered muscle mitochondrial function [20]. Notably, the development of kidney disease itself has been linked to the loss of renal insulin sensitivity [21,22] and to mitochondrial changes in multiple renal cell types [23,24]. 

In this mini review, we discuss how renal mitochondrial activity changes in DKD and explore the bioenergetics of skeletal muscle defects that may result from kidney disease. With respect to the function and dysfunction of mitochondria, we focus on their involvement in ATP synthesis, redox biology and oxidative stress. Effective engagement of mitochondria in these processes is influenced by their structural integrity, but mechanisms that ensure such integrity by regulating the dynamic morphology of mitochondria will not be discussed here. They have been recently reviewed expertly by others in context of nutrient metabolism [25], kidney disease [23,26] and the maintenance of muscle mass [27].

## 2. Renal Mitochondrial Activity in Diabetic Kidney Disease

Hyperglycaemia is the most important risk factor for kidney disease in people with diabetes [11], which explains why CKD is a prevalent microvascular complication of both T1D [28] and T2D [29]. Additional risk factors include hypertension, dyslipidaemia, insulin resistance and obesity [11], which are all features of the metabolic syndrome that increases the likelihood of developing T2D [30]. Renal function depends on the interplay between multiple cell types [31] and demands much energy—the kidney’s specific resting metabolic rate is 440 kcal kg^−1^ day^−1^ [32]. Diabetic kidney disease is characterised by filtration defects that manifest as persistent and increasing presence of protein in the urine (albuminuria) and a progressive decline in the glomerular filtration rate that may end in complete loss of kidney function [33]. These clinical symptoms are preceded by more silent disease stages in which glomerular filtration is either unaffected or in fact increased [33]. However, morphological changes to the glomeruli can be detected early in DKD [34] and include thickening of the glomerular basement membrane, mesangial expansion, podocyte loss and hypertrophy of remaining podocytes. Renal lesions that arise at a later stage include glomerulosclerosis and tubulointerstitial fibrosis [34].

### 2.1. Bioenergetics

Tubular reabsorption demands high amounts of energy, as ATP-fuelled Na^+^-K^+^ pumps establish the ion gradients that are needed for active trans-epithelial transport [35]. Since the required ATP is provided predominantly through oxidative phosphorylation [36], kidney function relies heavily on mitochondria, and mitochondrial dysfunction is thus believed to be central to DKD [23]. Renal ATP demand is raised during early-stage DKD, as hyper-filtration (cf. [33]) necessitates additional tubular reabsorption. Increased ATP synthesis in the kidney [37] at the outset of DKD is indeed consistent with extra energy expenditure. Interestingly, the additional ATP synthesis seems accounted for by glycolysis [37], not oxidative phosphorylation, suggesting a shift from mitochondrial to glycolytic ATP supply [38]. Notably, increased oxygen uptake by the kidney in early DKD is not coupled to ATP synthesis [39]. Similarly, the increased respiration exhibited by podocytes following prolonged exposure to 30 mM glucose, is insensitive to oligomycin and is thus not used to produce ATP [40]. Uncoupling protein-2 (UCP2) has been held responsible for the diabetes-related uncoupled respiration in renal proximal tubular cells [41] and in mitochondria isolated from the kidney cortex [42]. However, since the molecular and physiological functions of UCP2 remain the subject of significant debate [43], the nature of UCP2 involvement in DKD is accordingly uncertain [42,44,45]. For instance, *Ucp2* gene polymorphisms have been linked to the glomerular filtration rate in T2D patients and to decreased expression of renal *Ucp2* [46], which is at odds with the asserted UCP2 involvement in uncoupled respiration [41].

As DKD progresses, renal ATP synthesis [47], glucose oxidation [48] and fatty acid oxidation [49] start to decline. Mitochondrial activity changes accordingly in primary human glomerular mesangial cells (HMCs) after a 12-day exposure to 25 mM glucose [50]. Real-time functional bioenergetic analysis has demonstrated that such exposure lowers both coupled and uncoupled respiration [50]. It thus transpires that hyperglycaemia diminishes both the total and spare mitochondrial respiratory capacities of HMCs [50]. Exposure to a high glucose concentration has similar effects on mitochondrial oxygen uptake by immortalized proximal tubular (HK-2) cells [51]. It is thus likely that hyperglycaemia impairs the mitochondrial ATP synthesis capacity of HMCs and HK-2 cells, although it is formally possible that this capacity is lowered in response to decreased ATP demand during the extended glucose exposure. Notably, decreased mitochondrial respiratory capacity coincides with increased glycolysis in HK-2 cells but not in HMCs [51]. This contrasting glycolytic response might relate to a dissimilar energy demand, but it is presently unclear if it reflects differences between primary and immortalized cells, or between mesangial and tubular cells. Consistent with the effects of glucose excess on kidney cells [50,51], peripheral blood mononuclear cells (PBMCs) from patients with DKD have a comparably low spare respiratory capacity [50]. It has thus been suggested that systemic mitochondrial dysfunction underpins development of DKD [50]. It seems indeed likely that a diabetic milieu affects multiple bodily compartments, but the causal interrelations between the respective bioenergetic changes remain unclear.

### 2.2. Redox Biology and Oxidative Stress

The ATP synthesis rate of most cell types adjusts to ATP demand [16]. When metabolic fuel supply outweighs energy expenditure, many redox enzymes attain a reduction potential that allows partial reduction of oxygen [52] and thus formation of reactive oxygen species (ROS). For instance, at least 11 distinct mitochondrial enzymes are capable of producing ROS under reducing conditions [53]. These ROS have important physiological functions, but cause damage when they accumulate to excess. This duality is exemplified in skeletal muscle, where hydrogen peroxide is essential for insulin signalling, but causes oxidative stress and insulin resistance at persistently high concentrations [54]. Oxidative stress has been linked to diabetic complications [55], and a unifying pathological mechanism identifies mitochondria as source of the responsible ROS [56,57]. Intuitively, it seems likely that chronic hyperglycaemia disturbs the bioenergetic balance between renal energy demand and metabolic fuel supply, thus favouring ROS generation, but it remains contentious whether the ROS originate from mitochondria [58] or arise elsewhere, for example from activity of cytosolic NADPH oxidases [59]. Irrespective of the precise origin and molecular nature of the culprit, ROS-induced damage is indeed evident in diabetes, as urinary oxidised guanosine (a marker of oxidative DNA damage) correlates with mortality in T2D patients with albuminuria [60], and increases the risk of kidney disease in T1D patients [61]. In apparent contrast, kidneys of mice with streptozotocin-induced type 1 diabetes exhibit a *lower* ROS production rate—measured as in vivo oxidation of systemically administered dihydroethidium—than their healthy counterparts [48]. Nevertheless, glomerular oxidised guanosine and renal mitochondrial DNA (mtDNA) mutations that were detected in the diabetic mice [48] suggest a history of oxidative stress. Pharmacological stimulation of AMP-activated protein kinase (AMPK) increases renal ROS in these models [48], which is unexpected, as AMPK triggers antioxidant defence mechanisms [62]. Equally surprisingly, the increased ROS production coincides with a decreased renal mtDNA mutation frequency and a decreased level of glomerular oxidised guanosine [48]. 

Despite the strong association between DKD and oxidative damage, the therapeutic potential of antioxidants is uncertain. Meta-analyses of published clinical trials suggest positive effects [63,64], but benefits may be limited to early-stage DKD [63] and could perhaps be enhanced by prolonged administration [64]. It is unclear if added antioxidants are able to reach the ROS sources, which complicates the interpretation of disappointing trial results. Avoiding the accessibility issue, stimulation of endogenous antioxidant defences by pharmacological activation of *Nrf2* with bardoxolone methyl, may indeed improve kidney function [65], but the cardiotoxicity of this drug has stopped its clinical use [66]. When the antioxidant co-enzyme Q10 (ubiquinone) is targeted to mitochondria through conjugation to a triphenylphosphonium moiety, it protects against DKD in diabetic *db*/*db* mice [67]. Renoprotection offered by this MitoQ compound appears to arise from a mitochondrial uncoupling effect, not an antioxidant one [67]. Notably, uncoupling of oxidative phosphorylation will tip the cellular bioenergetic balance to energy expenditure, thus establishing an oxidised environment that disfavours ROS formation. Lack of positive clinical trial results may also reflect possible involvement of ROS with kidney physiology, which will likely be disturbed by antioxidants. It is worth emphasising in this respect that kidney function is regulated by insulin [68], and that renal insulin sensitivity is lost early in DKD [21,22]. Although speculation, antioxidants may inadvertently increase the risk of renal failure if ROS were equally important for insulin signalling in kidney as in skeletal muscle [54].

An alternative take on the disappointing clinical antioxidant trials is based on mitochondrial hormesis [69]. Mitohormesis is defined as an adaptive cellular response to mild mitochondrial stresses that render cells more resilient when exposed to subsequent, more severe, episodes of the same stress [70]. In the case of mild oxidative stress, mitochondrial ROS are thought to trigger an endogenous antioxidant defence that offers long-lasting protection against more intense future oxidative stress. Added antioxidants are expected to interfere with this hormetic response, as exemplified by their adverse influence on the health-promoting, ROS-mediated, effects of physical exercise [71]. It is conceivable that kidney cells safeguard themselves against oxidative stress by mitohormesis, and that added antioxidants compromise this endogenous defence mechanism. However, chronic administration of antioxidants should be able to compensate for an inappropriate endogenous response to oxidative stress, as indeed indicated by clinical trials [64]. In any case, the suggested potential of promoting mitochondrial superoxide production in DKD therapy [69] is not clear, as possible mitohormesis would have been compromised because of exogenous antioxidants, not because of an intrinsic inability of renal cells to produce ROS. Even experimental DKD models that exhibit relatively low ROS levels show signs of oxidative damage, i.e., renal mtDNA mutations and glomerular oxidised guanosine [48]. In this context, it is worth emphasising that a high mitochondrial respiratory activity does not necessarily imply a high rate of ROS production. As noted above, it is the reduction potential of redox enzymes that dictates ROS formation [52]. This potential is at least partly set by the balance between nutrient supply and energy expenditure. If driven by a high ATP demand, respiration will likely cause limited ROS production.

## 3. Skeletal Muscle Insulin Resistance

Loss of systemic insulin sensitivity is an important feature of the metabolic syndrome, and obesity-related insulin resistance of skeletal muscle and the liver has been studied intensively. Although pathological mechanisms are incompletely understood, supraphysiological fatty acid levels are broadly accepted culprits, and changes in mitochondrial activity are widely recognised [2]. Irrespective of obesity, skeletal muscle can also lose insulin sensitivity as a consequence of kidney failure. In this section, we argue that this uraemic muscle insulin resistance may arise against a background of imbalanced cellular bioenergetics similar to that more commonly associated with obesity (cf. [16]).

### 3.1. Obesity

The causality of mitochondrial involvement in nutrient-induced insulin resistance has been debated for some time [72,73], and the dispute is ongoing [74]. There is direct evidence from studies involving humans with congenital insulin receptor signalling defects, that lowered mitochondrial ATP synthesis capacity can follow from insulin resistance [75]. As we have discussed in detail recently [16], it is less evident to what extent mitochondrial respiratory dysfunction causes loss of insulin sensitivity. Insufficient mitochondrial capability to deal with nutrient excess in obesity has been suggested to produce lipid metabolites and ROS that interfere adversely with insulin signalling paths (Figure 2). Against this notion, it has been reasoned that muscle has a considerable spare oxidative capacity, in both lean and obese subjects, that should suffice to burn large nutrient loads fully. However, ATP fluxes are predominantly demand-driven in muscle and other tissues, thus engagement of the mitochondrial oxidative capacity depends on cellular energy demand. Indeed, mitochondrial models of obesity-related insulin resistance all involve harmful metabolites that only accumulate when nutrient supply exceeds energy expenditure [16]. It is thus crucial to evaluate mitochondrial dysfunction in the context of ATP consumption. When the energy demand of different experimental models is taken into account, many discrepancies in the literature are thus readily reconciled [76].

Regarding the ‘mitochondrial causality’ debate, it is worth noticing that a low mitochondrial ATP synthesis rate does not necessarily reflect an intrinsic oxidative phosphorylation defect, but could be an adaptation to altered energy demand, as obese subjects tend to be less physically active than lean individuals. Moreover, cellular stress responses that are induced by nutrient excess and pro-inflammatory cytokines will likely change energy demand. Although stress responses may be expected to increase (not decrease) mitochondrial ATP synthesis activity, it is conceivable that it perturbs the hierarchy of ATP-consuming processes [77]. For instance, less ATP may be allocated to processes that underpin general cell maintenance, or such processes may indeed be directly compromised by nutrient excess. Consistently, we have demonstrated that palmitate and stearate lower the rate of de novo protein synthesis in cultured rat myocytes, which coincides with a decreased rate and efficiency of oxidative ATP synthesis and with attenuation of insulin-sensitive glucose uptake [78]. The mitochondrial and glucose uptake effects of fatty acids are also observed in human myocytes [78]. Moreover, cycloheximide-sensitivity of coupled mitochondrial respiration is lowered in both human and rat muscle cells, which demonstrates that the energy demand linked to protein synthesis is decreased [78]. Notably, insulin resistance *per se* may lower energy demand, as insulin will be less able to stimulate ATP-consuming anabolic processes.

### 3.2. Uraemia

It has been recognised for some time that kidney disease can cause loss of skeletal muscle insulin sensitivity [79,80]. This uraemic muscle insulin resistance emerges early in CKD [81] and indeed manifests itself in patients with inherited forms of the disease before symptoms of renal failure are apparent [82]. Although the pathological mechanism is presently unclear, the insulin resistance has been attributed to changes in insulin signalling that occur downstream from the insulin receptor [83,84,85] in the insulin receptor substrate (IRS), phosphatidylinositol-3-kinase (PI3K) and protein kinase B (AKT) pathway [86]. The same signalling components are thus affected in uraemia as in obesity (Figure 2). Effectors that alter insulin signalling in CKD have not been established conclusively [17], but may well be similar to the culprits behind nutrient-induced skeletal muscle insulin resistance (Figure 2), i.e., lipid metabolites and ROS. This may seem counter-intuitive when considering that uraemic muscle insulin resistance manifests itself irrespectively of obesity [82] and that renal failure tends to lower fat mass [87]. However, changes in lipid metabolism seen in CKD do resemble those observed in obesity, including a raised level of circulating free fatty acids [88]. Moreover, macrophage infiltration of adipose tissue in lean CKD patients causes inflammation that is similar to that seen in obese subjects with normal kidney function [89,90]. This inflammation contributes to chronic oxidative stress in uraemia [91,92], and both inflammation and oxidative stress have indeed been implicated in the loss of muscle insulin sensitivity in CKD [17]. Additional consequences of renal failure that can cause muscle insulin resistance include a perturbed blood pH regulation [93,94] and accumulation of toxins because of impaired renal clearance [95]. These uraemic retention molecules have attracted interest in particular as aggravators of CKD and its comorbidities [96,97]. For instance, p-cresyl sulphate is associated with loss of insulin sensitivity of cultured myotubes, as it interferes with the phosphorylation of IRS1 [98]. Precursors of p-cresyl sulphate and related toxins such as indoxyl sulphate are formed during amino acid metabolism in intestinal bacteria, and their accumulation is thus influenced by the composition of the gut microbiome, which changes in CKD [99]. Notably, the toxicity of these compounds arises from the inflammation and oxidative stress they cause in target tissues [100,101]. Similarly, urea that builds up in CKD is believed to cause insulin resistance because of oxidative stress [102].

When produced excessively in obesity, hydrogen peroxide causes muscle insulin resistance via (indirect) inhibition of IRS1 [54,103]. Generally, formation of this ROS is only pathologically high when nutrient supply exceeds energy demand [16]. Such imbalanced bioenergetics are indeed characteristic of obesity, but are not necessarily a feature of CKD patients, many of whom are non-obese. However, the loss of adipose mass in kidney failure [87] coincides with accumulation of ectopic fat in many tissues, including skeletal muscle [98,104,105]. The balance between fuel supply and energy expenditure may thus well be perturbed at the cellular level in CKD, and thus permit high ROS formation. It is furthermore conceivable that a perturbed cellular bioenergetic balance causes build-up of lipid metabolites that lower insulin signalling. In this respect, mitochondrial insufficiency has been suggested to preclude complete lipid oxidation and thus cause uraemic skeletal muscle insulin resistance [99,106]. Decreased muscle mitochondrial respiratory capacity has indeed been reported in patients with CKD [20,90,107] and in models of kidney disease [90,108,109], but has not been linked directly to insulin resistance. As discussed in the context of obesity [16], low mitochondrial respiration may thus be a consequence of lost insulin sensitivity in uraemia rather than a cause.

## 4. Skeletal Muscle Wasting

The quality of the life of many CKD patients is compromised by frailty, particularly during the later stages of renal failure [110]. This frailty emerges at least partly because of changes in skeletal muscle mass and function that, collectively, are labelled as uraemic myopathy [111] or sarcopenia [112], the latter to reflect the progressive and cumulative nature of CKD effects on skeletal muscle [112]. Indeed, the muscle of CKD patients becomes increasingly weak as the disease progresses, in terms of power output, endurance as well as exercise tolerance [112]. Moreover, muscle tissue mass is lost because of a decreased size (atrophy) and number (hypoplasia) of muscle fibres [112]. General inactivity of CKD patients [113] does not fully explain this muscle weakness and muscle wasting, as both locomotory and non-locomotory muscles are affected [114]. Consistent with the very early manifestation of insulin resistance in CKD [81] and with the cumulative nature of its deleterious effects [112], defected insulin signalling has been implicated in the aetiology of uraemic myopathy [85,86,115]. This puts conditions that dampen insulin sensitivity (cf. Section 3.2) firmly in the mechanistic frame, and oxidative stress [111], inflammation [116], metabolic acidosis [117] and toxic uraemic retention molecules [101] have indeed all been linked to uraemic myopathy. Notably, lost insulin sensitivity in uraemia perturbs skeletal muscle proteostasis, as it upsets the balance between protein synthesis and protein breakdown [5,118]. Muscle proteolytic flux produces glucogenic amino acids used to fuel gluconeogenesis in the liver and kidney [119]. To maintain skeletal muscle mass, this protein turnover is compensated by de novo protein synthesis [120]. Defects in the IRS1-PI3K-AKT insulin signalling pathway (Figure 2) have been suggested [5] to lower protein synthesis and to permit proteolysis by activating the Forkhead box O (FOXO) transcription factors that regulate expression of genes involved in proteasome- and lysosome-mediated protein breakdown (Figure 1).

Protein synthesis [121] and protein breakdown [122] are both energy-demanding processes. It is thus perhaps not surprising that perturbed proteostasis has been linked to mitochondrial dysfunction. Compromised skeletal muscle bioenergetics have been associated with uraemic frailty and myopathy for some time, as it has been shown that the phosphocreatine recovery time after exercise is longer in the skeletal muscle of end-stage CKD patients than in healthy subjects, suggesting a relatively low ATP synthesis capacity [123,124,125]. Consistently, the activity of various energy metabolic enzymes is comparably low in the muscle of CKD patients [126,127,128,129], as is the myocellular mitochondrial volume density and mtDNA copy number [130]. Deleterious uraemic effects on muscle mitochondrial mass have also been observed in CKD mouse models [109], and in both clonal and primary mouse myotubes exposed to serum from such models [131]. Exposure to uraemic serum also attenuates mitochondrial biogenesis and ATP synthesis in cultured myotubes [131]. Decreased mitochondrial mass furthermore associates with increased autophagy, both in CKD patients [130] and mice [131]. Concomitant with a decreasing mitochondrial mass of muscle cells, PBMCs from patients with CKD have a decreased mtDNA content [130,132]. As in DKD, mitochondrial dysfunction thus appears to be a systemic aspect of CKD (cf. Section 2.1). Notably, however, DKD is associated with an increased (not decreased) level of circulating mtDNA that appears non-functional [50]. The decline in the mtDNA concentration of PBMCs in uraemia is proportional to the severity of kidney failure [130], and thus suggests progressive loss of mitochondrial activity as CKD develops. Indicative of causal involvement in uraemic myopathy, muscle mitochondrial density decreases before symptoms of muscle weakness emerge [130]. Non-invasive spectroscopic measurements of hand skeletal muscle bioenergetics in non-diabetic CKD patients with preserved physical performance have shown increased muscle oxygen uptake that is not used to make ATP [133]. It is unclear if this uncoupled mitochondrial respiration is an early pathological feature of CKD [133] that triggers a mitochondrial density decline, or that it is specific to muscle type.

The similarity of insulin signalling defects in uraemic and obese milieus (Figure 2) raises the question as to why skeletal muscle wasting seems more prominent in CKD than in obesity. This apparent discrepancy may just reflect the notion that insulin resistance is not the sole requirement for perturbed proteostasis in CKD (cf. [134]). Metabolic acidosis is, for example, a feature of uraemia that not only impairs insulin signalling [93,94], but also directly inhibits protein synthesis and directly stimulates proteolysis [117]. However, the relatively prominent muscle wasting in CKD could be interpreted differently. For instance, it is conceivable that pancreatic beta cells are less able to compensate for insulin resistance in CKD (and other muscle-wasting states) than in obesity [9], perhaps because of harmful effects of uraemic retention molecules on the insulin secretion from pancreatic beta cells [95]. Retention molecules that impair beta cell activity have indeed been reported and include 3-carboxy-4-methyl-5-propyl-2-furanopropanoic acid (CMPF) [135,136] as well as urea [137]. Notably, CMPF has been suggested to predict whether patients with prediabetes progress to T2D [138], which offers the possibility that the propensity of people with obesity-related insulin resistance to develop T2D is dictated by their kidney function.

## 5. Conclusions

Multiple aspects of the metabolic syndrome have been associated with bioenergetic failure, and mitochondria are, possibly, attractive therapeutic targets to treat metabolic disease. The reports we have reviewed above indicate two main mitochondria-related observations that are common to the pathologies we have discussed. Firstly, it appears that the oxidative capacity of skeletal muscle is decreased in obesity (cf. [16]) and in CKD [20,90,107,108,109]. Similarly, the mitochondrial respiratory capacity of different kidney cell types (and PBMCs) is lowered in DKD [50,51]. Under pathological conditions, in which respiration is actually increased—for example in kidney and skeletal muscle at the onset of DKD and CKD, respectively—oxygen consumption appears to have been uncoupled from ATP synthesis [39,133], which may be related to oxidative stress. Pharmacological strategies that boost oxidative phosphorylation may thus have therapeutic potential. It will be important to confirm, however, that the generally decreased ATP synthesis capacity is indeed due to intrinsic mitochondrial defects, as it cannot be excluded that mitochondrial ATP supply declines in response to pathological changes in energy demand. Secondly, it transpires that oxidative stress features in the loss of insulin sensitivity of both the muscle and kidney. Obesity is characterised by imbalanced bioenergetics that raise ROS generation sufficiently to impair muscle insulin signalling [54]. DKD associates with oxidative stress [55], and it is thus plausible that renal insulin resistance is an early consequence of the excessive ROS production in the kidney caused by hyperglycaemia. Similarly, ROS may play a causative role in the loss of muscle insulin sensitivity in CKD, since most factors that prevail in the uraemic milieu also induce oxidative stress [17]. Antioxidant therapies may be successful in preventing insulin resistance and linked pathologies, but the molecular nature and cellular origin of the harmful ROS need to be confirmed to allow specific drug targeting. Moreover, possible involvement of ROS in insulin signalling—and other physiological processes—needs to be firmly established to rationalise antioxidant strategies. 

## Figures and Tables

**Figure 1 ijms-20-02751-f001:**
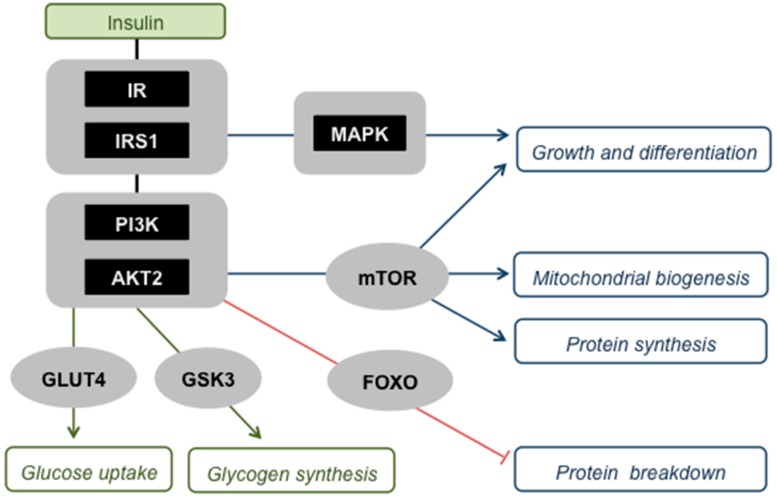
Skeletal muscle insulin signalling. Insulin activates its receptor (IR) and receptor substrate (IRS1), which comprise a ‘critical node’ in a branched signalling network that allows interaction with other pathways, for example those induced by cytokines [15]. Activation of this node triggers two major protein kinase cascades, i.e., the phosphatidylinositol-3-kinase (PI3K)—protein kinase B (AKT2 in skeletal muscle) pathway and the Ras-mitogen-activated protein kinase (MAPK) pathway, which both instruct muscle cells to engage with anabolic processes. Activated AKT2 has multiple effects: (i) It stimulates recruitment of the glucose transporter protein (GLUT4) to the plasma membrane, and is thus responsible for insulin-sensitive glucose uptake by muscle; (ii) it activates glycogen synthesis by inhibiting glycogen synthase kinase-3 (GSK3); (iii) it promotes mitochondrial biogenesis, protein synthesis, and cell growth and differentiation, effects that are all mediated through the stimulation of the mammalian target of rapamycin (mTOR); (iv) it suppresses protein breakdown by phosphorylating and thus deactivating Forkhead box O (FOXO) transcription factors that stimulate proteasome- and lysosome-mediated proteolysis. The MAPK pathway acts in concert with AKT2 to transmit insulin’s message to increase cell growth and differentiation.

**Figure 2 ijms-20-02751-f002:**
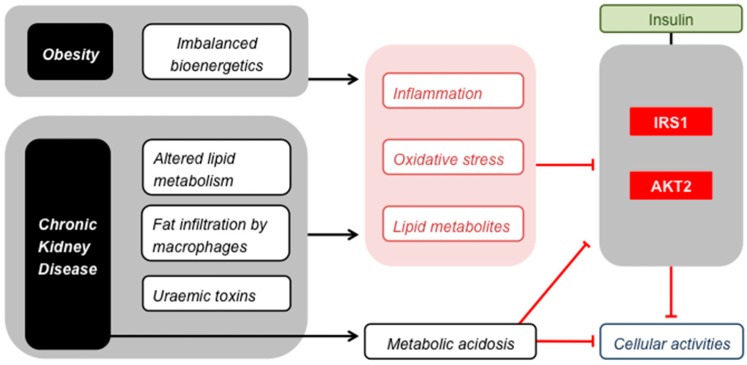
Skeletal muscle insulin resistance. Obesity and CKD cause common stresses that impair insulin signalling by inhibiting IRS1 and AKT2, and thus attenuate the effect of insulin on the cellular activities listed in Figure 1. Metabolic acidosis affects insulin signalling in a similar way, but is restricted to CKD.

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
