# Peer review of "Mitochondrial Activity and Skeletal Muscle Insulin Resistance in Kidney Disease"

_ijms, 2019, doi:10.3390/ijms20112751_

Round 1
Reviewer 1 Report
The authors focus on a scientifically important area that links the changes in kidney cell types during DKD development and discusses the controversial effects of oxidative stress and mitochondrial reactive oxygen species in DKD. The study fits in the scope of International Journal of Molecular Sciences. The manuscript is clearly written and well referenced. Each section is relevant and logically covers the topic. However, I have some suggestions to improve the manuscript: Comments: While it is becoming increasingly clear that mitochondrial dysfunction is involved in the development and progression of various kidney diseases leading to DKD. But the author does not seem to be interested in the quality and dynamics of mitochondria in this disease. As such, our understanding of the effects of mitochondrial dysfunction in DKD patients remains limited. In addition, the author's comprehensive assessment of mitochondrial function should help to describe the value and limitations of treatment strategies for restoring mitochondrial function in DKD patients.Author Response
The authors focus on a scientifically important area that links the changes in kidney cell types during DKD development and discusses the controversial effects of oxidative stress and mitochondrial reactive oxygen species in DKD. The study fits in the scope of International Journal of Molecular Sciences. The manuscript is clearly written and well referenced. Each section is relevant and logically covers the topic.
We are pleased with this positive reception of our manuscript.
However, I have some suggestions to improve the manuscript: Comments: While it is becoming increasingly clear that mitochondrial dysfunction is involved in the development and progression of various kidney diseases leading to DKD. But the author does not seem to be interested in the quality and dynamics of mitochondria in this disease. As such, our understanding of the effects of mitochondrial dysfunction in DKD patients remains limited.
We apologise if we have inadvertently given the impression that we are not interested in the structural integrity and dynamic morphology of mitochondria. We are interested, and indeed aware of the processes that safeguard structural integrity and regulate mitochondrial dynamics, but have chosen to focus our mini-review on mitochondrial activity measurements. We have added the following sentence to the end of the Introduction referring the reader to appropriate reviews on these topics:
'Effective engagement of mitochondria in these processes is influenced by their structural integrity, but mechanisms that ensure such integrity by regulating the dynamic morphology of mitochondria will not be discussed here. They have been recently reviewed expertly by others in context of nutrient metabolism [25], kidney disease [23,26] and maintenance of muscle mass [27].'
In addition, the author's comprehensive assessment of mitochondrial function should help to describe the value and limitations of treatment strategies for restoring mitochondrial function in DKD patients
Respectfully, we believe that we have indeed indicated the possible value and caveats of mitochondria as therapeutic targets, albeit concisely, in the Conclusion section of our manuscript.
Reviewer 2 Report
This is an interesting and well written manuscript, and I just want to mention two minor points:
Abstract: I suggest to better explain why you need to "evaluate how the mitochondrial activity of different renal cell types changes during development of DKD..." (lines 17-19), as now is not clear why we expect to see a link between mitochondrial activity and DKD.
Section No 2: I think than an excerpt from a textbook (lines 80-90) could be deleted, as the presented details are not necessary for the discussion.
Author Response
This is an interesting and well written manuscript, and I just want to mention two minor points:
We are pleased with this positive reception of our manuscript.
Abstract: I suggest to better explain why you need to "evaluate how the mitochondrial activity of different renal cell types changes during development of DKD..." (lines 17-19), as now is not clear why we expect to see a link between mitochondrial activity and DKD.
Re-reading the abstract, the focus on mitochondria indeed appears to come somewhat out of the blue. To more logically link the background information to the focus of our mini-review, we have inserted the following sentence:
'Bioenergetic failure has been associated with the loss of skeletal muscle insulin sensitivity in obesity and uraemia, as well as in the development of kidney disease and its sarcopenic complications.'
Section No 2: I think than an excerpt from a textbook (lines 80-90) could be deleted, as the presented details are not necessary for the discussion.
We agree and have cut this excerpt to the following statement:
'Renal function depends on the interplay between multiple cell types [31] and demands much energy – the kidney’s specific resting metabolic rate is 440 kcal kg-1 day-1 [32].'